# Exploring the Nutritional Potential of Wild Grass Fodder for Mega Herbivore (*Elephas maximus*) in the Foothills of Western Ghats

**DOI:** 10.3390/ani12192668

**Published:** 2022-10-04

**Authors:** Mohan Packialakshmi, Muthusamy Palani Divya, Krishnamoorthy Baranidharan, Seshadri Geetha, Kalipatty Nalliappan Ganesan, Manickam Vijayabhama, Srinivasan Manivasakan, Palanivel Hemalatha, Palaniswamy Radha, Meenakshisundaram Tilak, Venugopal Priyanka, Settu Krishnamoorthi, Balasubramaniam Vinothini, Jayesh Yuvraj Zende, Nikhil Balu Rajput

**Affiliations:** 1Department of Forest Products and Wildlife, Forest College and Research Institute, Tamil Nadu Agricultural University, Mettupalayam, Coimbatore 641301, Tamil Nadu, India; 2Dry Land Agricultural Research Station, Tamil Nadu Agricultural University, Chettinad, Sivagangai 630102, Tamil Nadu, India; 3Department of Pulses, Tamil Nadu Agricultural University, Coimbatore 641003, Tamil Nadu, India; 4Department of Forage Crops, Tamil Nadu Agricultural University, Coimbatore 641003, Tamil Nadu, India; 5Department of Basics and Social Science, Forest College and Research Institute, Tamil Nadu Agricultural University Mettupalayam, Coimbatore 641301, Tamil Nadu, India; 6Department of Agroforestry, Forest College and Research Institute, Tamil Nadu Agricultural University, Mettupalayam, Coimbatore 641301, Tamil Nadu, India; 7Department of Forest Biology and Tree Improvement, Forest College and Research Institute, Tamil Nadu Agricultural University, Mettupalayam, Coimbatore 641301, Tamil Nadu, India; 8Forest College and Research Institute, Tamil Nadu Agricultural University, Mettupalayam, Coimbatore 641301, Tamil Nadu, India; 9Department of Wildlife Sciences, College of Forestry, Vellanikara, Kerala Agricultural University, Thrissur 680656, Kerala, India

**Keywords:** native, grass fodder, nutritional character, elephant

## Abstract

**Simple Summary:**

The aim of the study is to improve elephant habitats by restoring them with wild grass fodder. Based on the feeding behaviour and food spectrum of elephants, the study documented 30 grass fodders. We used standard protocols to assess the nutritional analysis, and finally the study identified five nutrient rich potential grass fodder, viz., *Cynodon dactylon*, *Dichanthium aristatum*, *Heteropogon contortus*, *Oplismenus burmannii* and *Themeda triandra* for fodder bank development in corridors and fringe areas to improve the habitat of elephants. Hence, the findings are crucial and can be utilized for the management and conservation of elephantsin the Coimbatore Elephant Reserve (CER).

**Abstract:**

An elephant, being a mega herbivore, consumes large amounts of food. Due to the lack of availability of fodder inside the forest, the elephants move out of their habitat areas and also find agricultural crops attractive, which further results in man–animal conflict. To improve the elephant habitat area, the current study was conducted to assess the availability of native fodder grasses inside the Coimbatore Elephant Reserve, Western Ghats, from April 2021–April 2022. The area falls between 10°37′and 11°31′ North latitudes and 76°39′and 77°5′ East longitudes. It was approached in a systematic random sampling method. A total of 128 sample plots of 1 sq.m size were randomly placed, and the density of grass species was recorded in percentage (%). The collected samples were shade dried for one week, ground to pass through a 1 mm sieve, and stored in polythene bags. Furthermore, the samples were chemically analyzed to determine their nutritional values. The dry matter (DM) content of various grass fodder varied from 28.18% to 59.75%. The crude protein (CP) content differed between 5.94% and 11.94%. The highest CP was recorded in *Cynodon dactylon* (11.94%) and the least in *Aristida setacea* (5.94%). Ether extract content was found in the ranges of 1.00% to 5.00%. The acid detergent fibre (ADF) content of *Aristida setacea* (45.74%) was observed as the highest, whereas the lowest was observed in *Oplismenus burmannii* (26.78%), followed by *Themeda triandra* (26.85%), *Heteropogon contortus* (30.12%) and *Enteropogon monostachyos* (30.31%). The average neutral detergent fibre content of grass fodder was 52.27%, with a range of 37.89% (*Oplismenus burmannii*) to 67.87% (*Cymbopogon martinii*). The average total digestible nutrient (TDN) content of grass was 77.45%; relative forage quality (RFQ) exhibited wider variations among the grasses and ranged between 107.51 and 198.83. This study is a pioneer in evaluating the nutritional values of native grass fodder species for elephants in the Western Ghats. The study gives strategies for the selection of high nutritive fodder grass for the habitat improvement of elephants, and it also provides scientific and baseline information for the conservation of native grass fodder species in the Western Ghats.

## 1. Introduction

Elephants are mega herbivores and generalist foragers with a diverse diet which consists of grasses, forbs, fruits, bark, leaves, twigs and roots [1]. Owing to the unique morphology and physiology that accompanies their enormous body size, energy intake by elephants is very high, which is, however, constrained by their rate of forage quality [2]. The elephant consumes a large amount of food, estimated to be 1.5–2.5 percent of its body weight in dry weight fodder [3]. Adult elephants can consume approximately 300 kg of food per day. They can spend an average of 16–18 h per day on consumption. Due to this enormous need for food, elephants cannot afford to be highly selective feeders. In order to meet their fodder requirements, elephants move an average distance of 40 to 50 km, fixing their home ranges, and they follow these fixed routes annually every season [2,4]. Unfortunately, the umbrella species of the forest face a severe crisis due to the limitation of food in the forest territory areas. These factors drive the elephants from the forests to the associated fringe villages in search of food. A paradigm shift from conventional cropping to cash crops such as bananas and corns, often referred to as tempting crops for elephants, and its proximity to the elephant corridors, further attract theses gentle giants to come out of the forests. According to the reports of the WWF, India, in 2017, it wasestimated that approximately 301 elephants and 245 people are killed annually due to human–elephant conflicts in India. In addition, 500,000 families are affected every year by crop destruction and other human–elephant conflict-related issues [5]. Hence, habitat management is critical in adopting elephant conservation strategies in the Western Ghats. Incorporating ideal fodder species into the elephant habitat is one of the key elements in elephant habitat management. The seasonal movement and habitat selection of elephants are largely determined by their foraging behaviour [1]. Reports reveal that elephants in southern India intake heavy graminoids during the wet season [1],with it constituting 84.6% of their diet [6]. Grasses form a natural homogenous group of plants with remarkable diversity, and they play a significant role in the lives of wild animals [7]. They play a crucial role in the maintenance of the world’s ecosystems and biodiversity [8]. Studies on wild grasses, especially their fodder value, have become very important recently for the restoration of degraded ecosystems. Herbivores’ food resource selection is highly variable, and it can be influenced by the nutritional and energetic properties of the food plants. Henceforth, a deep understanding of the nutritional and energetic properties of the food plants consumed by elephants is essential to comprehend the feeding pattern and the selection of fodder plants by Asian elephants [9]. This alsoplays a crucial role in fostering habitat management strategies for elephant conservation in different phytogeographical regions of the Western Ghats. With this background, the present study has been conceived to explore the diversity of grasses around the Coimbatore Elephant Reserve and to evaluate their nutritional value.

## 2. Materials and Methods

**Study area:** The present study was conducted in the Mettupalayam and Sirumugai ranges of Coimbatore Forest Division, Western Ghats, Tamil Nadu, India, from April 2021–April 2022. The area falls between 10°37′and 11°31′ North latitudes and 76°39′and 77°5′ East longitudes. A greater part of the division is situated southwards in the Western Ghats, with the north-western parts forming the lower ranges of the Nilgiris. The elephant habitat area represents 20,000 ha of the study area, which was approached in a systematic random sampling method. A transact line of 2 km length was marked in the study area for exploration and documentation of grasses. For that, a 1 sq.m bamboo frame was randomly placed, and the density of grass species was recorded as apercentage (%) [10]. A sampling intensity of 0.2% was used. Sample plots (1 m×1 m) were laid in the opposite direction, and the distance between the sample plots were fixed as 200 m and 50 m from the transect line. A total of 128 sample plots was laid out in 6 beats of the Mettupalayam range and Sirumugai range, viz., Jaccanari, Sundapatti, Nellimalai, Hulikal, Kandiyur, Kallar, Odanthurai, Kunjapanai, Pethikuttai, Koothamundi north, Koothamundi south and Uliyur beat. The respective GPS points were recorded using a Gramin 60 version GPS. By using the geo-referencing points, the plots were marked on the Google Earth Map by using Google Earth Pro software, version 7.3 (Figure 1). Thesamplesof grass species collected from the plots were identified using their local names, whereas their scientific names were identified with the help of a book named “A handbook of Some South Indian grasses” [11]. Based on the survey and direct and indirect evidence of the feeding behaviour of elephants (Figure 2), 30 grass fodder species were catalogued, and the samples were collected for nutritional analysis.

Proximate analysis: The grass fodder samples were collected from the forest and were cut into small pieces so as to facilitate easy handling and uniform sampling for analysis. Samples were shade dried, and the dried samples were then ground to pass through a 1 mm sieve and were stored in polythene bags. The samples were chemically analyzed with three replications to determine their dry matter (DM), crude protein (CP), crude fibre (CF), ether extract (EE) and ash content (AC) using the AOAC [12] method; acid detergent fibre (ADF) and neutral detergent fibre (NDF) were determined as per Van Soest [13]; total digestible nutrient (TDN) and metabolizable energy (ME) as per Moran [14]; digestible dry matter (DDM) and dry matter intake (DMI) calculated as perSchroeder [15]; and relative forage quality (RFQ) calculated as perMoore and Undersander [16].

Statistical analysis: To compare the treatment means, the Tukey test was performed using the Statistical Package for Social Science data (SPSS) software, and the graphs were formed using Paleontological statistical software (PAST) version 3.

## 3. Results

The DM content of various grass fodder for elephant feeding varied from 28.18 ± 1.66% to 59.75 ± 4.00%. Most of the samples contained over 30% dry matter. The highest dry matter values were observed in *Arisida setacea* (59.75 ± 4.00%), followed by *Eragrostis cilianensis* (59.15 ± 2.25%), *Alloteropsis cimicina* (57.35 ± 2.64%) and *Perotis indica* (57.35 ± 2.49%). The ash content was found to be in the range of 1.60 ± 0.09% (*Eremochloa ophiuroides)* to 8.85 ± 0.18% (*Themeda triandra)*. It was observed to be highest in *Cynodon dactylon* (11.94 ± 0.80%), followed by *Themeda trinadra* (11.81 ± 0.93%), *Heteropogon contortus* (11.69 ± 0.75%), *Dichanthium aristatum* (11.56 ± 0.49%) and the lowestin *Aritistida setacea* (5.94 ± 0.22%). The average crude protein content of the grass was 8.11%. Ether extract content was found in the range of 1.00 ± 0.00% to 5.00 ± 0.31%. The crude protein content varied between 5.94 ± 0.22% (*Aristida setaceae*) and 11.94 ± 0.80% (*Cynodon dactylon*). The highest EE was observed in *Cymbopogon martini* (5.00 ± 0.31%), followed by *Enteropogon monostachyos* (4.67 ± 0.37%), *Heteropogon contortus* (4.33 ± 0.29%) and *Hyparrhenia hirta* (4.33 ± 0.17%) (Table 1).

The maximum crude fibre content was recorded in *Themeda triandra* (34.63 ± 2.95%) and it was followed by *Heteropogon contortus* (34.38 ± 1.20%) Table 2. The maximum ADF content was observed in *Aristida setacea* (45.74 ± 0.25%), whereas the minimum was observed in *Oplismenus burmannii* (26.78 ± 1.75%), followed by *Themeda triandra* (26.85 ± 2.36%), *Heteropogon contortus* (30.12 ± 0.91%) and *Enteropogon monostachyos* (30.31 ± 0.13%). The average NDF content of 30 grass fodder species was 52.27% with a range of 37.89 ± 2.07% (*Oplismenus burmannii*) to 67.87 ± 5.23% (*Cymbopogon martinii*), excluding *Cymbopogon martini* (67.87%) and *Eragrostis cilianensis* (60.16%) (Table 2).

Forage quality: Based on the primary constituents of nutritional attributes, the derived quality parameters, viz*.,* nitrogen free extract, total digestible nutrients, digestible dry matter, dry matter intake, metabolizable energy and relative forage quality were calculated. The total digestible nutrient (TDN) content was significantly different among the species, and it ranged between 62.21 ± 5.60% (*Themeda triandra*) and 87.26 ± 4.66% (*Digitaria longifolia*). Metabolizableenergy (ME) varied significantly among the grass fodder species and ranged from 9.62 ± 0.74 MJ/Kg DM to 14.25 ± 1.11MJ/Kg DM. In the current study, the RFQ exhibited wider variations among the grasses and ranged between 107.51 ± 6.94 and 198.83 ± 7.73. RFQ was recorded in *Perotis indica* (198.83 ± 7.73), followed by *Oplismenus burmannii* (192.77 ± 4.11) and *Kyllinga*
*brevifolia* (190.50 ± 11.49), whereas *Cymbopogon martinii* (107.51 ± 6.94) registered a minimum forage quality index (Table 3).

## 4. Discussion

Grasses were the most preferred feed resource by elephants in the study area, most prominently during the rainy season [4]. Dry matter (DM) is the actual amount of feed material excluding water, volatile acid and bases. The DM content varies according to the plant species, parts of the plant and its maturity during various growing conditions such as thesoil and environment. This can be attributed to the fact that the grass grows naturally in forest areas with adverse climatic conditions. This is in accordance with the findings of Khanum [17], who reported that the DM content of salt tolerant grasses varied from 31.30% (*Leptochloa fusca*) to 56.10% (*Cynodon dactylon*). The average ash content of the samples was 4.27%. This result is consistent with the results reported by Hamid et al. [18], who stated that the values of *Cenchrus ciliaris* were between the range of 8.98% and 9.14%. The CP values of the grass fodder species were observed to be above the critical value of 7.5% which is reported to be required for proper digestion (Figure 3). This was in accordance with the findings of Faji et al., Gate et al. and Abebe et al. [19,20,21],where the CP values were obtained in the range of 5.04% (Desho grass) to 6.98% (*Setaria sphacelata*), 6.13% to 9.63% (Baja × Napier hybrid) and 7.24% (*Cenchrus ciliaris*) to 8.90% (*Chrysopogon aucheri*). Conversely, it was in contrast with the findings of Adebayo et al. [22],who reported thatthe values of CP were from 9.49% to 25.86% in Guinea grass.

The study also revealed that the crude fibre content was found in the range from 10.67 ± 0.78% to 34.63 ± 2.95%. This might be attributed to the site quality, season, plant species and their growth pattern. The range of the crude fibres in the grasses studied is in accordancewith the findings of Karbivska et al. and Khude and Al-Rowaily et al. [8,23,24], who reported that the crude fibre content varied from 29.00% (*Lolium perenne*) to 30.40% (*Dactylis glomerata*), 25.00% (*Leptochola fusca*) to 28.50% (*Pennisetum purpureum*), and 11.97% (*Cyperus conglomeratus*) to13.84% (*Panicum turgidum*) respectively. The^.^ NDF content of all the grasses lies below the critical value of 60%. High NDF content in the fodder affects the voluntary feed intake and feed conversion efficiency. According to Singh and Oosting [25], if the roughage contains above 65% NDF, it is considered as poor quality feed. This supports the findings of Faji et al., Gate et al. and Megersa et al. [20,26,27], who reported that the ADF and NDF contents were in the range of 35.33% (*Panicum aquatica*) to 42.03% (*Panicum coloratum*) and 64.89% (*Panicum aquatica*) to 71.62% (*Chloris gayana*); 29.80% to 52.80% (Bajra x Naiper hybrid) and 61.40% to 77.60% (Bajra x Naiper hybrid); and 42.33% (*Pennisetum unssetm*) to 54.99% (*Pennisetum* sp.) and 72.45% (*Eustachys paspaloidsi*), respectively (Figure 4).

The average TDN content of grass fodder was 77.45% (Table 3). A similar result was reported by Hamid et al. [18], who reported that the values were in the range of 53.14% to 63.65% TDN, which provides an assessment of the energy content of the feed. The larger the value of TDN, the more energy is condensed within feedstuff. The study was in accordancewith those of Zewdu, Khude, and Bamikole and Ikhatua [8,28,29], where the values were found in the range of 8.61 MJ/Kg DM to 9.77 MJ/Kg DM (*Pennisetum purpureum*); 6.36 MJ/Kg DM (*Cynodon dactylon*) to 7.29 MJ/Kg DM (*Sporobolus arabicus*); and 8.77 MJ/Kg DM (*Pannicum maximum*). Among the forage quality parameters, RFQ is very essential because forage quality index is used to allocate forages to the herbivores with the given levels of expected performance [30]. DDM denotes the total digestibility of the feed, whereas DMI denotes the amount of feed an animal consumes as a percentage of its body weight and is calculated from its NDF percentage. The maximum RFQ was recorded in *Perotis indica*, followed by *Oplismenus burmannii*. This might be due to the lower ADF and NDF content in the above grass species than in other grass species, because the RFQ is derived from ADF and NDF [31] (Figure 5). The results were indirectly proportional to the ADF and NDF content (Figure 6). The study was in agreement with the findings of Faji et al. [26],who recorded the RFQ values in the range between 115.07% (*Chloris gyana*) and 122.92% (*Urochloa decumbens*).

## 5. Conclusions

Wild grass fodder species plays a vital role in improving herbivore productivity in natural forests, and it has potentially good protein supplements, particularly during the critical periods of the year when the quantity and quality of herbage are limited. The major limiting factor for an elephant’s biorhythm is feed, both in terms of quantity as well as quality. To curb the problem of feed shortage, the incorporation of wild grass fodder species could be regarded as a one-stop solution in habitat management. The unavailability of feed sources is one of the most significant concerns for elephant conservation in the Western Ghats, and can be solved through the rehabilitation of elephant corridors and habitat areas. In order to rehabilitate these elephant corridors, authenticated and consistent information on the nutritional value of native fodder grass species is required. This study is highly significant for being the pioneer research on the nutritional evaluation of native grass fodder species in the Western Ghats, which can form the backbone for management plans to enrich the habitats of elephant corridors. The identified nutrient rich grass fodder species, viz., *Cynodon dactylon, Dichanthium aristatum, Heteropogon contortus, Oplismenus burmannii* and *Themeda triandra,* are highly recommended for fodder bank development in corridors and fringe areas to cater to the needs of the megaherbivore and its niche areas.

## Figures and Tables

**Figure 1 animals-12-02668-f001:**
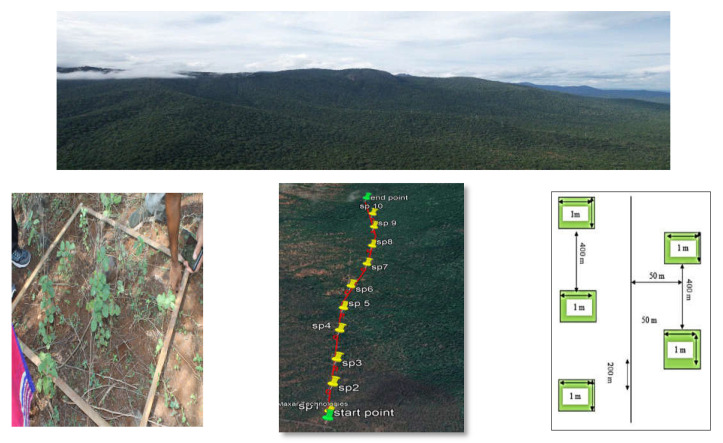
Map shows 1 × 1 m size of sample plots for grass fodder in the study area. (SP- Sample plot numbers).

**Figure 2 animals-12-02668-f002:**
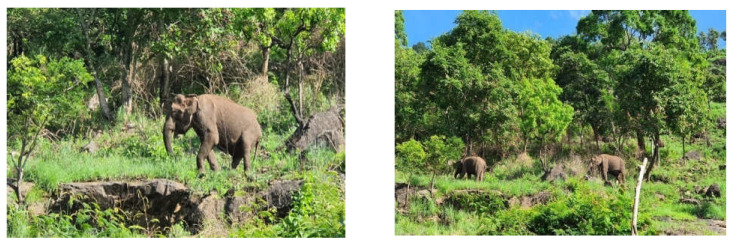
Direct feeding behaviour (grass fodder) of elephants in the study area.

**Figure 3 animals-12-02668-f003:**
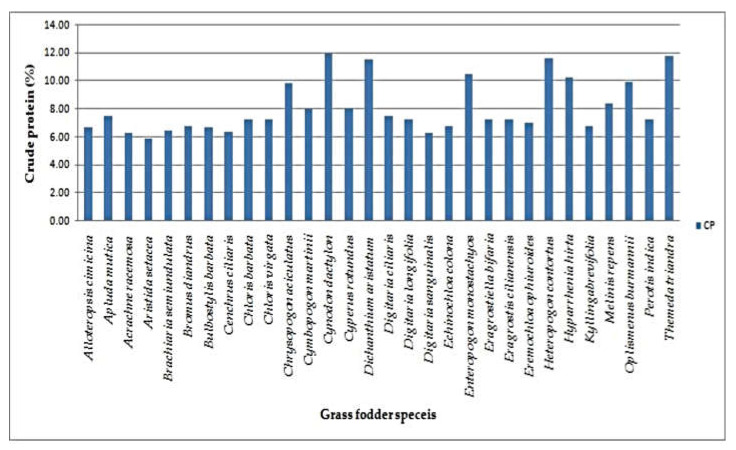
Crude protein content of native grass fodder species.

**Figure 4 animals-12-02668-f004:**
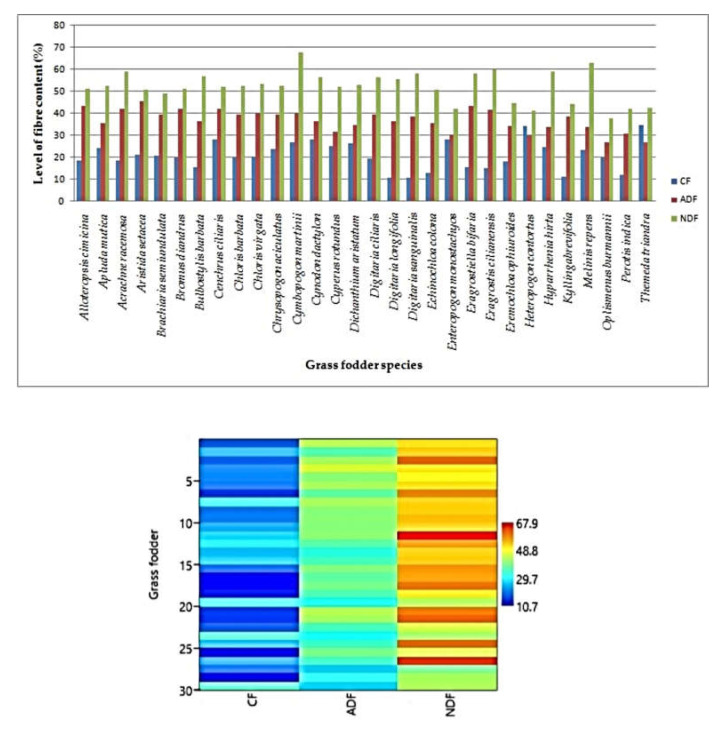
Crude fibre (CF), acid detergent fibre (ADF) and neutral detergent fibre (NDF) per cent in native grass fodder species.

**Figure 5 animals-12-02668-f005:**
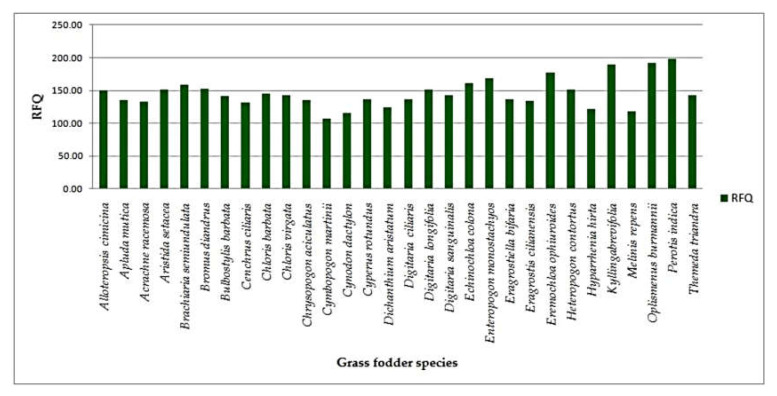
Relative forage quality of native grass fodder species.

**Figure 6 animals-12-02668-f006:**
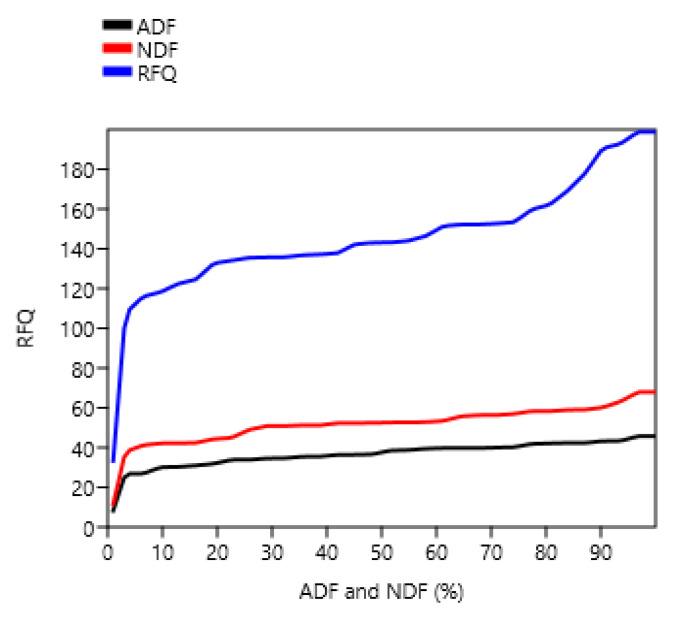
Relationship between detergent fibres and relative feed quality.

**Table 1 animals-12-02668-t001:** Nutritional value of native grass fodder species (%).

	Species Name	DM	AC	CP	EE
1.	*Alloteropsis cimicina*	57.35 ± 2.64^abc^	4.25 ± 0.27^fg^	6.75 ± 0.19^fg^	1.67 ± 0.15^gh^
2.	*Apluda mutica*	47.79 ± 1.07^efgh^	7.10 ± 0.01^cd^	7.50 ± 0.37^cdef^	3.00 ± 0.05^c^
3.	*Acrachne racemosa*	40.26 ± 1.09^ijkl^	3.15 ± 0.21^ij^	6.31 ± 0.37^fg^	2.67 ± 0.14^de^
4.	*Aristida setacea*	59.75 ± 4.00^a^	3.00 ± 0.12^jk^	5.94 ± 0.22^g^	2.00 ± 0.01^fg^
5.	*Brachiaria semiundulata*	54.01 ± 4.62^abcde^	1.85 ± 0.14^lm^	6.44 ± 0.15^fg^	1.33 ± 0.02^hi^
6.	*Bromus diandrus*	53.80 ± 1.03^abcdef^	2.35 ± 0.14^l^	6.81 ± 0.34^efg^	2.33 ± 0.19^ef^
7.	*Bulbostylis barbata*	33.33 ± 1.11^lmno^	2.40 ± 0.06^kl^	6.75 ± 0.18^fg^	1.33 ± 0.11^hi^
8.	*Cenchrus ciliaris*	41.27 ± 1.05^hijk^	6.90 ± 0.02^d^	6.38 ± 0.45^fg^	2.67 ± 0.13^de^
9.	*Chloris barbata*	50.75 ± 3.91^cdef^	3.35 ± 0.20^hij^	7.31 ± 0.36^cdef^	1.00 ± 0.08^i^
10.	*Chloris virgata*	48.48 ± 0.48^defg^	3.45 ± 0.18^hij^	7.25 ± 0.36^cdef^	1.00 ± 0.00^i^
11.	*Chrysopogon aciculatus*	55.07 ± 3.01^abcd^	5.10 ± 0.24^e^	9.88 ± 0.15^b^	1.33 ± 0.06^hi^
12.	*Cymbopogon martinii*	46.63 ± 1.47^fghi^	4.35 ± 0.32^fg^	8.00 ± 0.14^cde^	5.00 ± 0.31^a^
13.	*Cynodon dactylon*	47.09 ± 0.44^efgh^	7.90 ± 0.24^b^	11.94 ± 0.80^a^	2.33 ± 0.18^ef^
14.	*Cyperus rotundus*	32.21 ± 2.40^mno^	4.60 ± 0.36^ef^	8.06 ± 0.41^cd^	1.67 ± 0.07^gh^
15.	*Dichanthium aristatum*	53.37 ± 0.71^abcdef^	8.70 ± 0.72^a^	11.56 ± 0.49^a^	1.67 ± 0.09^gh^
16.	*Digitaria ciliaris*	34.16 ± 1.77^jklmno^	4.20 ± 0.04^fg^	7.56 ± 0.54^cdef^	2.33 ± 0.09^ef^
17.	*Digitaria longifolia*	37.13 ± 2.44^jklmn^	2.05 ± 0.03^lm^	7.31 ± 0.12^cdef^	2.67 ± 0.03^de^
18.	*Digitaria sanguinalis*	44.48 ± 1.81^ghij^	3.00 ± 0.24_jk_	6.31 ± 0.15^fg^	1.33 ± 0.08^hi^
19.	*Echinochloa colona*	36.87 ± 1.17^jklmn^	3.35 ± 0.24^hij^	6.81 ± 0.21^efg^	2.67 ± 0.02^de^
20.	*Enteropogon monostachyos*	37.40 ± 2.58^ijklmn^	3.70 ± 0.21^ghi^	10.50 ± 0.19^b^	4.67 ± 0.37^ab^
21.	*Eragrostiella bifaria*	55.07 ± 2.23^abcd^	3.25 ± 0.05^ij^	7.25 ± 0.46^cdef^	1.33 ± 0.02^hi^
22.	*Eragrostis cilianensis*	59.15 ± 2.25^ab^	2.15 ± 0.06^lm^	7.31 ± 0.31^cdef^	1.67 ± 0.13 ^gh^
23.	*Eremochloa ophiuroides*	38.97 ± 3.49^jklm^	1.60 ± 0.09^m^	7.06 ± 0.15^defg^	1.00 ± 0.07^i^
24.	*Heteropogon contortus*	31.37 ± 1.96^no^	7.55 ± 0.37^bc^	11.69 ± 0.75^a^	4.33 ± 0.29^b^
25.	*Hyparrhenia hirta*	28.18 ± 1.66^o^	5.00 ± 0.30^e^	10.31 ± 0.01^b^	4.33 ± 0.17^b^
26.	*Kyllinga brevifolia*	52.51 ± 2.95^abcdef^	1.75 ± 0.13^lm^	6.81 ± 0.30^efg^	1.00 ± 0.06^i^
27.	*Melinis repens*	33.33 ± 1.71^lmno^	4.00 ± 0.33^fgh^	8.38 ± 0.53^c^	3.33 ± 0.16^c^
28.	*Oplismenus burmannii*	38.97 ± 0.57^jklm^	7.20 ± 0.15^cd^	9.94 ± 0.11^b^	3.00 ± 0.09^cd^
29.	*Perotis indica*	57.35 ± 2.49^abc^	2.15 ± 0.10^lm^	7.31 ± 0.63^cdef^	2.33 ± 0.14^ef^
30.	*Themeda triandra*	32.78 ± 0.76^mno^	8.85 ± 0.18^a^	11.81 ± 0.93^a^	3.00 ± 0.13^c^
	**SEM**	**1.83**	**0.19**	**0.33**	**0.12**

^a^ SEM:standard error of mean. DM: dry matter, AC: ash content; CP: crude protein; EE: ether extract. According to theTukey test, mean values with different superscript (abcdefghijklmno) within a column are significantly different (*p* < 0.05). The data are expressed as mean ± Standard deviation (SD).

**Table 2 animals-12-02668-t002:** Crude fibre and Detergent fibre components of the grass fodder species.

	Species Name	CF	ADF	NDF
1.	*Alloteropsis cimicina*	18.63 ± 1.01^ijk^	43.32 ± 1.03^gh^	51.23 ± 0.64^cdefg^
2.	*Apluda mutica*	24.29 ± 0.25^cdef^	35.45 ± 2.33^bcd^	52.65 ± 0.10^defgh^
3.	*Acrachne racemosa*	18.76 ± 0.77^ijk^	42.13 ± 2.89^fgh^	59.08 ± 0.85^ghi^
4.	*Aristida setacea*	21.38 ± 0.69^efghi^	45.74 ± 0.25^h^	50.87 ± 0.41^cdefg^
5.	*Brachiaria semiundulata*	20.88 ± 1.56^fghi^	39.76 ± 2.12^defgh^	48.97 ± 4.16^bcde^
6.	*Bromus diandrus*	20.13 ± 1.66^ghi^	42.34 ± 0.97^fgh^	51.23 ± 3.79^cdefg^
7.	*Bulbostylis barbata*	15.68 ± 1.29^jkl^	36.34 ± 2.09^bcdef^	56.98 ± 2.93^efghi^
8.	*Cenchrus ciliaris*	28.18 ± 1.36^bc^	42.34 ± 0.14^fgh^	52.31 ± 4.30^cdefgh^
9.	*Chloris barbata*	20.11 ± 0.02^gh^	39.76 ± 2.97^defgh^	52.43 ± 4.70^cdefgh^
10.	*Chloris virgata*	20.43 ± 0.52^fghi^	40.01 ± 2.93^defgh^	53.45 ± 0.14^efgh^
11.	*Chrysopogon aciculatus*	23.90 ± 1.17^defg^	39.76 ± 3.49^defgh^	52.65 ± 2.94^defgh^
12.	*Cymbopogon martinii*	26.86 ± 1.52^bcd^	40.21 ± 2.63^defgh^	67.87 ± 5.23^j^
13.	*Cynodon dactylon*	28.45 ± 1.40^b^	36.43 ± 1.97^bcdef^	56.32 ± 2.79^efghi^
14.	*Cyperus rotundus*	25.32 ± 1.78^bcde^	31.89 ± 1.02^abc^	52.31 ± 1.15^cdefgh^
15.	*Dichanthium aristatum*	26.33 ± 0.40^bcd^	34.65 ± 1.64^bcd^	52.87 ± 1.84^defgh^
16.	*Digitaria ciliaris*	19.35 ± 1.50^ij^	39.45 ± 1.89^defgh^	56.32 ± 4.30^efghi^
17.	*Digitaria longifolia*	10.67 ± 0.78^n^	36.65 ± 2.52^cdef^	55.76 ± 0.62^efghi^
18.	*Digitaria sanguinalis*	10.83 ± 0.74^n^	38.57 ± 0.40^defg^	58.34 ± 0.46^fghi^
19.	*Echinochloa colona*	13.26 ± 1.10^lmn^	35.53 ± 0.26^bcde^	50.84 ± 0.66^cdef^
20.	*Enteropogon monostachyos*	28.11 ± 2.34^bc^	30.31 ± 0.13^abc^	42.16 ± 0.42^ab^
21.	*Eragrostiella bifaria*	15.71 ± 0.96^jkl^	43.27 ± 1.09^gh^	58.34 ± 3.28^fghi^
22.	*Eragrostis cilianensis*	15.19 ± 1.16^klm^	41.89 ± 3.34^efgh^	60.16 ± 2.44^hij^
23.	*Eremochloa ophiuroides*	18.13 ± 0.42^ijk^	34.55 ± 2.64^bcd^	44.89 ± 0.81^abcd^
24.	*Heteropogon contortus*	34.38 ± 1.20^a^	30.12 ± 0.91^ab^	41.23 ± 0.08^ab^
25.	*Hyparrhenia hirta*	24.91 ± 1.00^bcde^	33.87 ± 1.13^bcd^	58.94 ± 2.29^fghi^
26.	*Kyllinga* *brevifolia*	11.31 ± 0.67^mn^	38.76 ± 2.89^defg^	44.31 ± 3.18^abc^
27.	*Melinis repens*	23.31 ± 0.95^defgh^	33.87 ± 1.18^bcd^	63.12 ± 2.08^ij^
28.	*Oplismenus burmannii*	19.91 ± 0.54^hi^	26.78 ± 1.75^a^	37.89 ± 2.07^a^
29.	*Perotis indica*	12.29 ± 0.13^lmn^	30.98 ± 0.39^abc^	42.14 ± 1.34^ab^
30.	*Themeda triandra*	34.63 ± 2.95^a^	26.85 ± 2.36^a^	42.42 ± 1.62^ab^
	**SEM**	**1.00**	**1.63**	**2.09**

^a^ All values represented are on a dry matter basis. ADF: acid detergent fibre, NDF: neutral detergent fibre. According to the Tukey test, mean values with different superscript (abcdefghijk) within a column are significantly different (*p* < 0.05). The data areexpressed as mean ± SD.

**Table 3 animals-12-02668-t003:** Forage quality parameters of grass fodder species.

	Species Name	NFE	TDN	DDM	DMI	ME	RFQ
1.	*Alloteropsis cimicina*	68.70 ± 1.01^bcde^	79.51 ± 1.07^abc^	55.15 ± 2.15^de^	2.34 ± 0.01^defg^	12.82 ± 0.12^abcdf^	151.42 ± 3.68^defg^
2.	*Apluda mutica*	58.11 ± 2.30^cfgh^	73.23 ± 5.86^abcde^	61.28 ± 4.95^abcde^	2.28 ± 0.03^efgh^	11.66 ± 0.50^dfgh^	135.69 ± 3.28^ghij^
3.	*Acrachne racemosa*	69.11 ± 0.33^abde^	81.19 ± 5.25^ab^	56.08 ± 0.17^cde^	2.03 ± 0.07^ghi^	13.13 ± 1.07^abcdf^	134.08 ± 11.18^ghij^
4.	*Aristida setacea*	67.68 ± 3.21^bcdef^	79.39 ± 5.62^abc^	53.27 ± 2.21^e^	2.36 ± 0.16_defg_	12.80 ± 0.69^abcdf^	152.25 ± 3.53^defg^
5.	*Brachiaria semiundulata*	69.50 ± 3.21^abcd^	80.20 ± 3.79^abc^	57.93 ± 3.65^bcde^	2.45 ± 0.14^cdef^	12.95 ± 0.91^abcdf^	159.79 ± 6.96^cdef^
6.	*Bromus diandrus*	68.38 ± 1.53^bcde^	80.56 ± 4.75^abc^	55.92 ± 2.25^cde^	2.34 ± 0.18^defg^	13.01 ± 0.65^abcdf^	153.43 ± 8.73^defg^
7.	*Bulbostylis barbata*	73.84 ± 2.00^abc^	83.10 ± 1.69^ab^	60.59 ± 4.27^abcde^	2.11 ± 0.04^fghi^	13.48 ± 1.12^abcd^	142.29 ± 1.10^efghi^
8.	*Cenchrus ciliaris*	55.87 ± 1.01^gh^	71.16 ± 1.65^bcde^	55.92 ± 0.60^cde^	2.29 ± 0.13^efh^	11.27 ± 0.17^fghi^	132.72 ± 1.62^ghij^
9.	*Chloris barbata*	68.23 ± 4.01^bcde^	78.70 ± 2.33^abc^	57.93 ± 3.80^bcde^	2.29 ± 0.14^efgh^	12.67 ± 0.60^abcdf^	146.45 ± 11.04^defgh^
10.	*Chloris virgata*	67.87 ± 3.29^bcde^	78.42 ± 1.05^abc^	57.73 ± 4.65^cde^	2.25 ± 0.15^fgh^	12.62 ± 0.04^abcdf^	143.14 ± 7.59^efghi^
11.	*Chrysopogon aciculatus*	59.79 ± 0.11^efg^	73.28 ± 3.92^abcde^	57.93 ± 2.92^bcde^	2.28 ± 0.06^efgh^	11.67 ± 0.44^dfgh^	135.78 ± 0.97^ghij^
12.	*Cymbopogon martinii*	55.79 ± 4.46^gh^	74.79 ± 6.49^abcde^	57.58 ± 0.41^cde^	1.77 ± 0.12^i^	11.95 ± 0.60^cdfg^	107.51 ± 6.94^k^
13.	*Cynodon dactylon*	49.38 ± 1.73^hi^	66.95 ± 1.66^cde^	60.52 ± 4.58^abcde^	2.13 ± 0.07^fghi^	10.50 ± 0.73^ghi^	115.97 ± 7.81^jk^
14.	*Cyperus rotundus*	60.35 ± 0.95^defg^	73.89 ± 3.60^abcde^	64.06 ± 1.68^abcd^	2.29 ± 0.03^efgh^	11.78 ± 0.34^dfgh^	137.82 ± 0.49^fghij^
15.	*Dichanthium aristatum*	51.74 ± 3.33^gh^	67.52 ± 1.67^cde^	61.91 ± 1.28^abcde^	2.27 ± 0.09^fgh^	10.60 ± 0.28^ghi^	124.60 ± 5.80^hijk^
16.	*Digitaria ciliaris*	66.56 ± 0.18^cdef^	78.97 ± 0.36^abc^	58.17 ± 5.11^abcde^	2.13 ± 0.01^fghi^	12.72 ± 0.36^abcdf^	136.80 ± 5.94^ghij^
17.	*Digitaria longifolia*	77.30 ± 0.89^ab^	87.26 ± 4.66	60.35 ± 3.06^abcde^	2.15 ± 0.15^fghi^	14.25 ± 1.11^a^	152.68 ± 9.15^defg^
18.	*Digitaria sanguinalis*	78.53 ± 5.30^a^	86.11 ± 7.44^a^	58.85 ± 0.76^abcde^	2.06 ± 0.04^fghi^	14.04 ± 0.08^a^	144.00 ± 10.05^efghi^
19.	*Echinochloa colona*	73.91 ± 0.73^abc^	84.53 ± 2.83^ab^	61.22 ± 0.20^abcde^	2.36 ± 0.13^defg^	13.75 ± 0.78^abc^	162.20 ± 10.34^cde^
20.	*Enteropogon monostachyos*	53.02 ± 2.58^gh^	73.06 ± 1.63^abcde^	65.29 ± 2.86^abc^	2.85 ± 0.14^abc^	11.63 ± 0.46^dfgh^	169.06 ± 9.24^bcd^
21.	*Eragrostiella bifaria*	72.46 ± 4.55^abc^	82.02 ± 4.25^ab^	55.19 ± 0.66^de^	2.06 ± 0.08^fghi^	13.28 ± 0.52^abcd^	137.17 ± 1.60^fghij^
22.	*Eragrostis cilianensis*	73.68 ± 2.20^abc^	83.55 ± 2.88^ab^	56.27 ± 1.37^cde^	1.99 ± 0.03^ghi^	13.57 ± 0.17^abc^	135.50 ± 5.12^ghij^
23.	*Eremochloa ophiuroides*	72.21 ± 5.46^abc^	81.84 ± 2.41^ab^	61.99 ± 0.73^abcde^	2.67 ± 0.05^bcde^	13.25 ± 0.11^abcd^	177.86 ± 8.48^abc^
24.	*Heteropogon contortus*	42.05 ± 1.92^i^	64.34 ± 0.94^de^	65.44 ± 1.35^abc^	2.91 ± 0.25^ab^	10.01 ± 0.19^hi^	152.25 ± 11.50^defg^
25.	*Hyparrhenia hirta*	55.45 ± 4.80^gh^	73.97 ± 2.15^abcde^	62.52 ± 2.02^abcde^	2.04 ± 0.18^ghi^	11.79 ± 0.48^dfgh^	122.44 ± 2.50^ijk^
26.	*Kyllinga brevifolia*	79.13 ± 2.28	86.52 ± 5.95^a^	58.71 ± 1.98^abcde^	2.71 ± 0.18^bcd^	14.12 ± 0.53^a^	190.50 ± 11.49^ab^
27.	*Melinis repens*	60.98 ± 4.83^defg^	76.51 ± 5.72^abcd^	62.52 ± 5.57^abcde^	1.90 ± 0.15^hi^	12.27 ± 0.67^bcdfg^	118.26 ± 6.50^jk^
28.	*Oplismenus burmannii*	59.95 ± 4.37^defg^	74.87 ± 6.25^abcde^	68.04 ± 5.62^a^	3.17 ± 0.01^a^	11.96 ± 0.64^cdfg^	192.77 ± 4.11^a^
29.	*Perotis indica*	75.92 ± 2.48^abc^	85.88 ± 7.58^ab^	64.77 ± 3.86^abcd^	2.85 ± 0.23^abc^	14.00 ± 0.44^a^	198.83 ± 7.73^a^
30.	*Themeda triandra*	41.71 ± 1.68^i^	62.21 ± 5.60^e^	67.98 ± 4.38^ab^	2.83 ± 0.09^abc^	9.62 ± 0.74^i^	143.08 ± 3.11^efghi^
	**SEM**	**2.45**	**3.47**	**2.56**	**0.10**	**0.49**	**5.82**

^a^ All values are represented on a dry matter basis; NFE: nitrogen free extract; TDN: total digestible nutrient; DDM: digestible dry matter; DMI: dry matter intake; ME: metabolizable energy; RFQ: relative forage quality; ^b^ According to the Tukey test, mean values with different superscript (abcdefghijkl) within a column are significantly different (*p* < 0.05). The data are expressed as mean±SD.; ^b^ Units of NFE, TDN, DMI and DDM are expressed in %. ^c^ Units of ME are represented as MJ/Kg DM.

## Data Availability

The data can be found within the article.

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
