# Peer review of "Exploring the Nutritional Potential of Wild Grass Fodder for Mega Herbivore (Elephas maximus) in the Foothills of Western Ghats"

_animals, 2022, doi:10.3390/ani12192668_

Round 1

Reviewer 1 Report

This is a novel study of the food resources used by a charismatic megaherbivore, the Asian elephant Elephas maximas. The data analyzed and presented here on the properties of food plants, such as dry matter and crude protein, will benefit researchers and land managers in understanding patterns of diet use by elephants, and will aid in the conservation and management of the elephants and their habitat. I am not aware of any previous studies that have examined the nutritional properties of the food resources of Asian elephants in India in such detail, and so this study presents a valuable new perspective on the interactions between a megaherbivore and the potential food resources. This study is therefore likely to be of interest to researchers and land managers.

Although there are many positive aspects, in my view the manuscript would benefit from some changes, which I have listed below. 

Specific comments:
Lines 20-23: The beginning of the abstract focuses on methodological detail, but it it not explained why the study is being conducted. Before the sentences on methodology, there must first be one or more sentences to explain the biological question being addressed in the study. Otherwise the reader cannot understand the rationale for why the study was carried out.

Line 24: The past tense of "grind" is "ground"; therefore, "grinded" should be amended to "ground".

Line 26: All abbreviations such as DM should be explained on their first use. Some readers may not be familiar with the abbreviation of DM for Dry Matter. This also applies to other abbreviations used in the abstract, such as CP, ADF, NDF, TDN, and RFQ.

Line 44: To improve the language, amend "required" to "requires".

Lines 45-46: An appropriate citation should be included here to support this statement.

Line 46: To improve the language, amend "elephant" to "elephants".

Line 48: I recommend adding a sentence here to describe the results of the study by Baskaran et al. (2010), who found that grasses comprised 84.6% of the diet of Asian elephants. Baskaran, N., et al. (2010). Feeding ecology of the Asian elephant Elephas maximus Linnaeus in the Nilgiri Biosphere Reserve, southern India. Journal of the Bombay Natural History Society, 107(1), 3-13. 

Line 49: "Foraging is a major factor in elephant movement and its habitat selection". This statement  should be supported an appropriate citation, such as Sukumar (1989), who found that the seasonal habitat preferences of Asian elephants were driven in part by changes in the palatability of their food plants: Sukumar, R. (1989). Ecology of the Asian elephant in southern India. I. Movement and habitat utilization patterns. Journal of Tropical Ecology, 5(1), 1-18.

Line 52: To improve the language, the phrase "cannot afford to the selective feeder" should be amended to "cannot afford to be a highly selective feeder".

Lines 52-54: "To meet its fodder requirement, elephant moving around 40 to 50 km and they have to fix their home range and follow these fixed routes every year in the same season." This statement should be supported by an appropriate citation, so that the reader understands the source of this information. For example: Sukumar, R. (2006). A brief review of the status, distribution and biology of wild Asian elephants Elephas maximus. International Zoo Yearbook, 40(1), 1-8.

Lines 64-65: "Corridor management and conservation requires authenticate and consistent information on grass fodder composition and their diversity pattern." The manuscript currently does not inform the reader why this information is required. You state that this information is needed but you do not explain why is it important to know the properties of the food plants. It would be informative to add additional text and appropriate supporting citations to explain why this information is required. For example, you could state that "Herbivore selection of food resources is highly variable, and can be influenced by the nutritional and energetic properties of the food plants (Wood et al. 2019). Therefore, to understand the use of fodder plants by Asian elephants, information on the nutritional and energetic properties of their food plants is required." Wood, K.A., et al. (2019). Seasonal variation in energy gain explains patterns of resource use by avian herbivores in an agricultural landscape: Insights from a mechanistic model. Ecological Modelling, 409, 108762.

Line 66: As you mention "human-elephant conflict mitigation", it would be informative to cite some examples of such conflicts, to provide the reader with appropriate background information. For example: Gubbi, S.,et al. (2014). An elephantine challenge: human–elephant conflict distribution in the largest Asian elephant population, southern India. Biodiversity and Conservation, 23(3), 633-647. Shaffer, L.J., et al. (2019). Human-elephant conflict: A review of current management strategies and future directions. Frontiers in Ecology and Evolution, 6, 235.

Lines 107-109: "Data generated from the experiment were analyzed using the General Linear model analysis of variance procedure of Statistical Package for Social Science data (SPSS)." It is not clear how General Linear Models were used in this context, and so more information needs to be reported. What were the dependent and independent variables in these analyses? How many GLMs were used? Were the assumptions of the GLMs met in all cases? If so, this should be stated; if not, then you should describe what steps you took to rectify this, i.e. data transformations. More information on the assumptions of GLMs can be found in this paper: Zuur, A.F., et al. (2010). A protocol for data exploration to avoid common statistical problems. Methods in Ecology and Evolution, 1(1), 3-14.

Lines 127-129: In the table information you should also define "SEd" and "CD". 

Lines 158-159: "Grasses are mostly preferred feed resources by elephant in the study area and mostly used during rainy season." This statement should be supported by an appropriate citation, e.g. Baskaran, N., et al. (2010). Feeding ecology of the Asian elephant Elephas maximus Linnaeus in the Nilgiri Biosphere Reserve, southern India. Journal of the Bombay Natural History Society, 107(1), 3-13. 

Line 173: In Figure 2, the CP values in the left graph should not be connected by a line, as the CP value of each plant is independent of the CP value of the other plant species. For this reason, a bar chart / histogram would be a more appropriate figure for the CP data.

Line 193: In Figure 3, the CF, ADF, and NDF values should not be connected, as the value for each plant is independent of the values of the other plant species. For this reason, a bar chart / histogram would be a more appropriate choice for the presentation of these data.

Line 211: For Figure 4, I have the same comment as for Figures 2 and 3. The values presented are independent of each other, and hence should not be connected by a line. For this reason, a bar chart / histogram would be a more appropriate choice for the presentation of these data.

Line 213: In the legend for Figure 5, you should indicate what each of the circles plotted in the figure represents (is it a distinct plant species?). What do the connecting lines represent?

Lines 217-219: "Corridor management and conservation requires authenticate and consistent information on nutritional value of native fodder grasses for rehabilitation of elephant habitat area in Coimbatore elephant reserve." This repeats some of the text already presented in the introduction (lines 64-65). The conclusion should offer a broader perspective on the study topic. For example, it would be more informative to state that "Accurate and robust information, of the type that we have provided here, has been shown to aid in the management of human-wildlife conflicts (Redpath et al. "2015)." Redpath, S.M., et al. (2015). Conflicts in conservation: navigating towards solutions. Cambridge University Press. 

Reviewer 2 Report

The research is vital for the region and other reasons you explained in the article. However, the statistical analysis and the whole paper must be improved to have a higher impact on the readers.

Round 2

Reviewer 2 Report

Although there are some improvements to the manuscript, it still needs more work to be published, mainly in terms of materials and methods, and discussion. The English writing must also be improved and the figures.
